# Loss of MBNL1 induces RNA misprocessing in the thymus and peripheral blood

Łukasz J. Sznajder [1,5,6✉], Marina M. Scotti[1,5], Jihae Shin[1,3], Katarzyna Taylor[1,4], Franjo Ivankovic [1], Curtis A. Nutter[1], Faaiq N. Aslam[1], S. H. Subramony[2], Laura P. W. Ranum[1] & Maurice S. Swanson [1,6✉]

The thymus is a primary lymphoid organ that plays an essential role in T lymphocyte maturation and selection during development of one arm of the mammalian adaptive immune response. Although transcriptional mechanisms have been well documented in thymocyte development, co-/post-transcriptional modifications are also important but have received less attention. Here we demonstrate that the RNA alternative splicing factor MBNL1, which is sequestered in nuclear RNA foci by C(C)UG microsatellite expansions in myotonic dystrophy (DM), is essential for normal thymus development and function. *Mbnl1* 129S1 knockout mice develop postnatal thymic hyperplasia with thymocyte accumulation. Transcriptome analysis indicates numerous gene expression and RNA mis-splicing events, including transcription factors from the TCF/LEF family. *CNBP*, the gene containing an intronic CCTG microsatellite expansion in DM type 2 (DM2), is coordinately expressed with *MBNL1* in the developing thymus and DM2 CCTG expansions induce similar transcriptome alterations in DM2 blood, which thus serve as disease-specific biomarkers.

[1] Department of Molecular Genetics and Microbiology, Center for NeuroGenetics and the Genetics Institute, University of Florida, College of Medicine, Gainesville, FL 32610, USA. [2] Department of Neurology, Center for NeuroGenetics, University of Florida, College of Medicine, Gainesville, FL 32610, USA. [3] Present address: Department of Microbiology, Biochemistry and Molecular Genetics, Rutgers New Jersey Medical School and Rutgers Cancer Institute of New Jersey, Newark, NJ 07103, USA. [4] Present address: Laboratory of Gene Therapy, Department of Gene Expression, Institute of Molecular Biology and Biotechnology, Faculty of Biology, Adam Mickiewicz University, Umultowska 89 61-614 Poznań, Poland. [5] These authors contributed equally: Łukasz J. Sznajder, Marina M. Scotti. [6] These authors jointly supervised this work: Łukasz J. Sznajder, Maurice S. Swanson. ✉email: lukasz.j.sznajder@gmail.com; mswanson@ufl.edu

The thymus plays a crucial role in adaptive immunity by providing a conducive environment for T lymphocyte (T cell) differentiation[1,2]. As thymocytes mature, approximately 90–95% undergo apoptosis, either from lack of effective T-Cell Receptor (TCR) recombination and signaling, or alternatively, in response to negative selection to ablate high-affinity TCRs capable of recognizing self-antigens[3]. Defects in T cell selection and/or the loss of mature T cell populations, can result in a compromised immune state where cytotoxic CD8$^+$ T cells are unable to exert cytotoxic functions against intracellular pathogens, or helper CD4$^+$ T cells are unable to effectively provide B cell help to drive protective humoral immunity. While transcriptional regulation has been shown to play a vital role in both thymic organogenesis and T cell development[4], co/post-transcriptional events have also been implicated in thymocyte differentiation[5,6]. Indeed, alternative splicing (AS) adds another layer of complexity and diversity for developmental and tissue-restricted expression and splicing patterns for many gene transcripts that shift during development and aging[7–9].

Due to the complexity of interactions between the RNA sequence elements and trans-acting splicing factors that control splicing and 3′-end cleavage/polyadenylation, RNA processing is particularly susceptible to mutations implicated in disease[10,11]. For example, the Muscleblind-like (MBNL) family of alternative splicing factors triggers the switch from fetal to adult splicing programs for its RNA targets and loss of MBNL1 activity is a major pathogenic factor in the multisystemic disease myotonic dystrophy (DM) type 1 (DM1) and type 2 (DM2)[12,13]. DM1 and DM2 are autosomal dominant disorders caused by microsatellite, or short tandem repeat (STR), CTG and CCTG expansions ($^{exp}$) encoded in either the 3′ untranslated region (UTR) of DMPK (DM1) or the first intron of CNBP (DM2), respectively. In both DM types, transcription of these mutant STRs results in the expression of C(C)UG$^{exp}$ RNAs that are retained in the nucleus as RNA foci together with MBNL proteins[14]. This MBNL sequestration process results in downstream pre-mRNA misprocessing, including errors in AS and alternative 3′-end cleavage/polyadenylation (APA)[15,16] that result in pathological manifestations[17].

Although DM1 and DM2 are classified as a muscular dystrophy, the immune system is also affected and both DM types are characterized by a number of cellular and humoral abnormalities in peripheral blood. For example, although hypogammaglobulinemia and low lymphocyte counts occur in both DM types, they are especially prevalent in DM2 and are associated with an increased risk of autoimmune disease in DM2[18,19]. In addition, thymic hyperplasia and thymoma, as well as increased risk for other cancer types, have been reported in DM[20–23]. Although the immune phenotype contributes to DM1 and DM2 complexity, the consequence of MBNL depletion on adaptive immunity has not been investigated.

The thymus is active in developing mice and highly active in the pre-pubescent period in humans, but subsequently undergoes progressive involution with reduced thymic output. In this study, we report that loss of MBNL1 expression in 129S1-$Mbnl1^{\Delta E3/\Delta E3}$ knockout (KO) mice results in postnatal thymic hyperplasia and thymocyte accumulation and we identify misprocessing of developmental splicing events critical for T lymphocyte maturation. Importantly, these splicing changes are also detectable in DM2, but not DM1, peripheral blood. Additionally, we provide evidence that the degree of splicing dysregulation is proportional to DM2 CCTG STR length and CNBP intron 1 retention level. Based on this analysis, we propose a set of AS events that are readily detectable in whole blood and serve as biomarkers for DM2 disease.

## Results

### MBNL1 loss leads to dysregulation of thymic gene expression.
Our previous in situ hybridization study of $Mbnl$ gene expression during mouse embryogenesis revealed that $Mbnl1$ is highly expressed in the thymus suggesting that the MBNL1 protein regulates RNA processing during thymic development[24]. To confirm this observation and extend our understanding of $Mbnl1$ developmental expression, we retrieved publicly available RNA sequencing (RNA-seq) data of embryonic (E12.5-E18.5) and newborn (P0) mouse thymus[25]. Differential gene expression analysis confirmed that $Mbnl1$ expression increased during thymic organogenesis with 5.7-fold higher expression at P0 compared to E12.5, and $Mbnl1$ was in the 99.4 percentile of expressed genes at P0 with 14− and 8−fold higher expression than $Mbnl2$ and $Mbnl3$, respectively (Fig. 1a and Supplementary Fig. 1a). In agreement with our prior in situ results, $Dmpk$ expression level was very low in the developing thymus in striking contrast to $Cnbp$ (Supplementary Fig. 1b).

During studies to assess genetic modifier effects on developmental regulation of RNA processing in the mouse $Mbnl1$ KO model of DM, B6.129S1-$Mbnl1^{\Delta E3/\Delta E3}$ mixed background mice were backcrossed to obtain either 129S1-$Mbnl1^{\Delta E3/\Delta E3}$ or B6-$Mbnl1^{\Delta E3/\Delta E3}$ N10 congenic lines. For subsequent studies, we selected 129S1-$Mbnl1^{\Delta E3/\Delta E3}$ mice, hereafter referred to as 129-$Mbnl1$ KO mice, due to their shortened lifespan compared to B6-$Mbnl1$ KOs with a median survival of 22 and 37 weeks of age, respectively (Fig. 1b). To determine if MBNL1 loss caused RNA misprocessing, thymi were isolated from 9-week-old (P63) 129-$Mbnl1$ KO and wild type (WT) littermates (Supplementary Fig. 1c). Paired-end (PE) RNA-seq demonstrated that $Mbnl1$ was in the 99.6 percentile of expressed genes in thymus whereas $Mbnl2$ and $Mbnl3$ expression remained 18- and 12-fold lower, respectively (Fig. 1a and Supplementary Fig. 1a). In agreement with previous studies on other cells and tissues, $Mbnl2$ expression increased 2-fold following MBNL1 loss (Supplementary Fig. 1d). Differential gene expression analysis revealed that ~5% of genes expressed in the 129-$Mbnl1$ KO thymus were misregulated (Fig. 1c). Of 1436 genes that showed expression changes in $Mbnl1$ KO thymus, 630 were upregulated while 806 were downregulated (Fig. 1c and Supplementary Fig. 1e) and 54% of these corresponded to expression differences that occur during embryonic development (Supplementary Fig. 1f). Interestingly, variations in the immunoglobulin heavy ($Ighv$) and kappa ($Igkv$) clusters were among the most downregulated genes (Fig. 1d). Because the TCR beta variable ($Tcrb$-$V$) region also showed misregulation (Supplementary Fig. 1g), we assessed both the Tcrb and Tcra repertoires in our RNA-seq dataset. Surprisingly, 2 out of 3 129-$Mbnl1$ KO thymi showed altered clonotype frequencies for both Tcrb and Tcra transcripts suggesting clonal expansion of thymocytes (Fig. 1e–g and Supplementary Fig. 1h). Since these results demonstrated that MBNL1 loss led to aberrant thymic gene expression and TCR alterations, we next investigated potential effects of these changes on thymus and thymocyte development.

### Thymic pathology in 129-$Mbnl1$ knockout mice.
Analysis of 129-$Mbnl1$ KO mice revealed a time-dependent and striking enlargement of the thymus (Fig. 2a and Supplementary Fig. 2a). Gross thymic morphology and weight in 129-$Mbnl1$ KOs were undetectable at 4–13 weeks of age, but by 19 weeks of age the KO thymus had significantly enlarged with normalized thymic wet mass weight increased from 3- to 88-fold (median 24-fold) compared to WT (Fig. 2b). Approximately 76% of males, and 40% of females, had an enlarged thymus characterized by lobulated or other irregular morphology. Histological assessment indicated massive thymocyte overgrowth especially in the thymic cortex

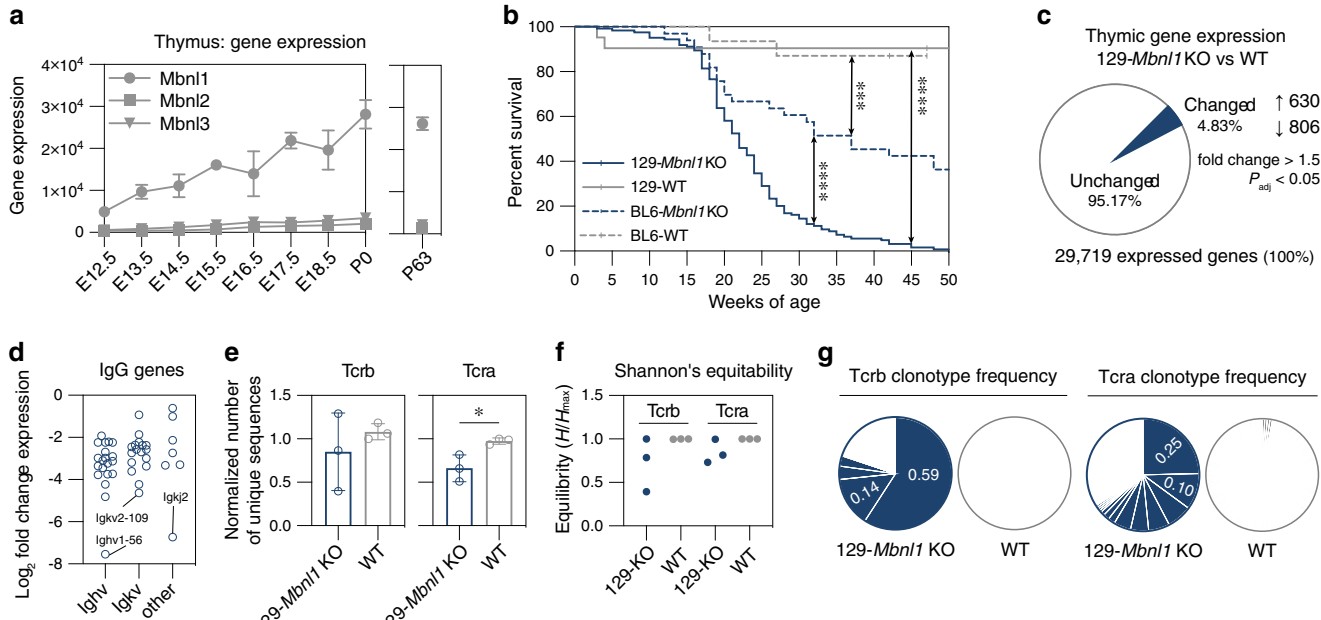

**Fig. 1 Mbnl1 regulates thymic development. a** *Mbnl1, Mbnl2* and *Mbnl3* gene expression levels during thymus organogenesis and in the developed gland. RNA-seq was performed at embryonic (E) days: 12.5 ($n = 3$), 13.5 ($n = 3$), 14.5 ($n = 2$), 15.5 ($n = 2$), 16.5 ($n = 3$), 17.5 ($n = 2$), 18.5 ($n = 2$) and postnatal days 0 (P0; $n = 2$) and 63 (P63; $n = 3$). Points connected by lines show mean expression ± standard deviation (SD). Data from embryonic and P0 thymi were obtained from GSE107910. **b** Survival curves for 129-*Mbnl1* knockout (KO) ($n = 124$, female/male ratio is 49/75), B6-*Mbnl1* KO ($n = 33$, 16/17), 129-wild type (WT) ($n = 21$, 7/14) and B6-WT ($n = 21$, 10/11). Significant differences between survival distributions were determined by Mantel–Cox (log-rank) test: *** $P < 0.001$; **** $P < 0.0001$. Median survivals are 22 and 37 weeks of age for 129-*Mbnl1* KO and B6-*Mbnl1* KO, respectively. There are no significant differences between females and males. **c** Gene expression changes in 129-*Mbnl1* KO thymus. Pie chart represents the proportion of significantly altered genes (blue) to all detected genes (gray) in 129-Mbnl1 KO ($n = 3$) compared to WT ($n = 3$) RNA-seq samples. **d** Scatter plot represents 44 downregulated genes encoding immunoglobulin heavy (*Ighv*; 20 genes), kappa (*Igkv*; 17 genes) and other (7 genes) chains. Each dot represents a significantly changed gene by DESeq2 analysis. The most affected gene for each group is indicated. **e** Reduction of the TCR repertoire detected by *Mbnl1* KO RNA-seq. Bar graph shows number of unique Tcrb and Tcra sequences normalized to the unique mapped read count ± SD. Significant difference was determined by two-tailed t-test: * $P = 0.027$. **f** Tcrb ($n = 3$) and Tcra ($n = 3$) repertoire diversity reflected by Shannon's equitability where 1 equals complete equivalency. **g** Pie charts demonstrate representative Tcrb and Tcra clonotype frequencies in 129-*Mbnl1* KO and WT thymic RNA-seq. Gray areas represent unique clones from the total clone count. Source data are provided as a Supplementary Data 1 file.

(Fig. 2c) and viable thymocyte counts revealed 4- to 30-fold increased cellularity in 129-*Mbnl1* KO enlarged thymi compared to WT (Fig. 2d). As expected, *Mbnl1* expression was ~5-times higher in developing thymocytes compared to medullary thymic epithelial cells (mTEC) (Supplementary Fig. 2b).

Mouse thymocytes progress from CD4$^-$CD8$^-$double negative (DN), through immature CD8$^+$ single positive (ISP), to CD4$^+$CD8$^+$ double positive (DP) developmental stages in the thymic cortex and become CD4$^+$ or CD8$^+$ single positive cells in the medulla[26] (Supplementary Fig. 2c). Flow cytometry highlighted an increased ratio of DN and ISP and/or mature CD8$^+$, as well as reduction of DP, cells (Fig. 2e, f and Supplementary Fig. 2d). Since ISP are larger than DP and mature CD8$^+$ thymocytes[26], we analyzed the size distribution of the CD8$^+$ subpopulation. The forward angle light scatter (FSC) analysis revealed an age-related increase in relative cell size distribution in 129-*Mbnl1* KO compare to WT indicating aberrant accumulation of ISP cells (Fig. 2g and Supplementary Fig. 2e). Interestingly, *Notch1* regulates maturation of mouse thymocytes and is highly expressed at both DN and ISP stages, and we observed an increase in Notch1 RNA in 9-week-old 129-*Mbnl1* KO thymi (Supplementary Fig. 2f)[27–29]. In agreement, significantly lower *Cd4* expression was detected by RNA-seq (Supplementary Fig. 2f). Finally, to extend our 129-*Mbnl1* KO characterization, we analyzed peripheral T cells in spleen, and noted an age-related

change in the CD4$^+$/CD8$^+$ ratio, possibly due to thymic dysfunction (Supplementary Fig. 2g, h).

Gene ontology analysis of differentially expressed genes in WT versus *Mbnl1* KO thymi revealed enrichment in multiple categories in the KO, including regulation of cell population proliferation (168 genes, FDR = $3 \times 10^{-10}$) and apoptotic process (124 genes; FDR = $4 \times 10^{-3}$) (Supplementary Data 1; sheet 21). Apoptosis is a critical aspect of thymocyte development, and cells that respond inappropriately, either too sensitive or insensitive, to apoptotic signals can cause a block in thymocyte development and accumulation of thymocytes[30]. Therefore, we tested the possibility that apoptosis was altered between 4 and ~22 (20 ± 2) weeks of age. Interestingly, *Mbnl1* KOs showed reduced sensitivity to dexamethasone-induced apoptosis with the greatest effect at 22 weeks of age (Supplementary Fig. 2i, j).

Cumulatively, these data suggested that aging 129-*Mbnl1* KO thymocytes can be retained in the thymus possibly due to an abnormal developmental process, including defective proliferation and/or apoptotic signaling. Although gene expression changes were detectable as early as 9 weeks of age when thymi were in the normal size range, 42% of 129-*Mbnl1* KOs did not develop enlarged thymi. Factors such as age, sex and genetic strain background impacted the penetrance of this phenotype.

Since MBNL1 is an RNA processing factor that regulates alternative splicing during postnatal development, we next

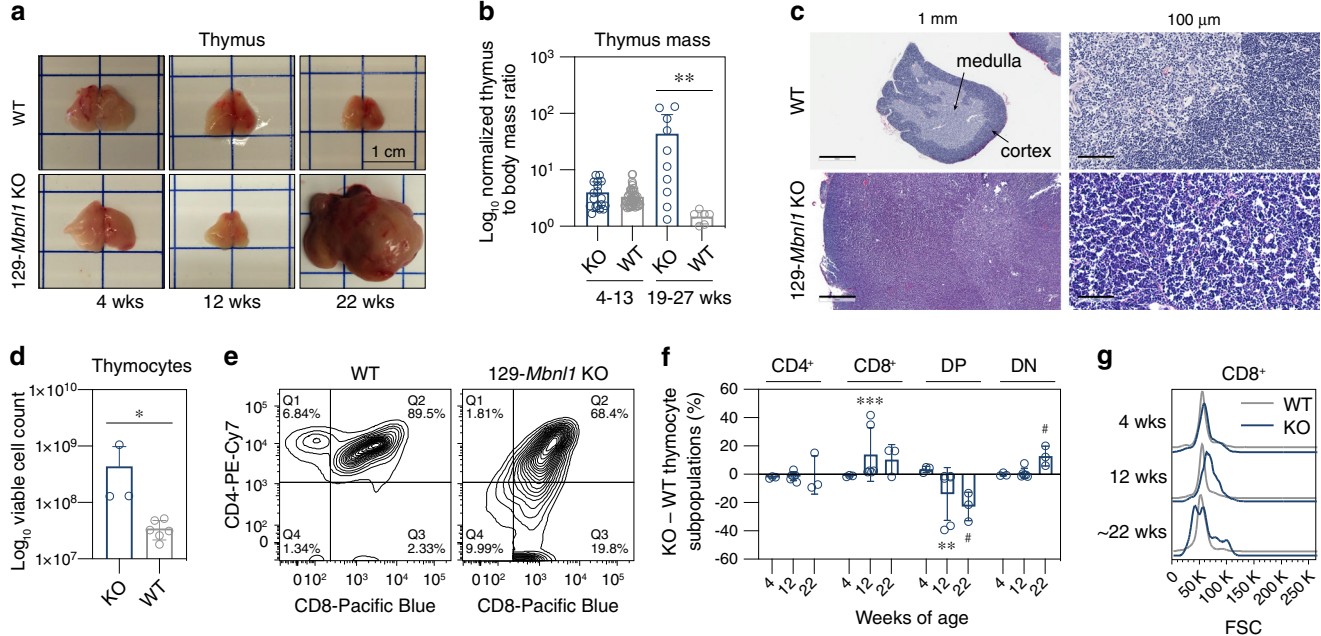

**Fig. 2 Thymic hyperplasia in aged Mbnl1 KO mice. a** Representative photos of male WT and 129-*Mbnl1* KO thymi at 4, 12, and 22 weeks of age. The normal process of thymic involution that occurs in WT is disrupted in 129-*Mbnl1* KO at 22 weeks. **b** Male thymic wet weight normalized to body weight is significantly increased in 129-*Mbnl1* KO ($n = 10$) compared to WT ($n = 6$) at 19–27 weeks. There is no significant difference between 4 and 13 weeks (WT = 50, KO = 18). Bar graph shows mean ± SD. Significant difference was determined by the Mann-Whitney test: \*\*$P = 0.003$. **c** Hematoxylin and eosin (H&E) stain of WT and enlarged 129-*Mbnl1* KO thymi. **d** Increased number of viable thymocytes in enlarged 129-*Mbnl* KO ($n = 3$) compared to WT ($n = 6$) thymi by ~22 (20± 2) weeks. Bar graph shows mean ± SD. Significant difference was determined by the Mann–Whitney test: \*$P = 0.024$. **e** Representative flow cytometry analysis of CD4 and CD8 expression in ~22 weeks WT and 129-*Mbnl1* KO thymocytes. **f** Thymocyte composition at 4 (WT = 3; KO = 3), 8 (WT = 8; KO = 6), and ~22 (WT = 4; KO = 3) weeks. CD4+, CD8+, CD4+CD8+ double positive (DP) and CD4-CD8- double negative (DN) thymocytes were analyzed by flow cytometry. Graph represents mean KO–WT subpopulations ± SD. Significant difference was determined by the Mann–Whitney test: # $P = 0.057$, \*\*$P = 0.007$, \*\*\*$P = 0 .0007$. **g** Representative flow cytometry Forward Scatter (FSC) parameter plots showing different relative thymocyte CD8+ cell sizes in 129-*Mbnl1* KO. Source data are provided as a Supplementary Data 1 file.

examined the 129-*Mbnl1* KO thymic transcriptome to test if RNA mis-splicing contributes to thymic pathology.

**Altered RNA splicing in *Mbnl1* knockout thymus.** To investigate the impact of MBNL1 deficiency on thymic pre-mRNA processing we performed alternative splicing (AS) analysis using our RNA-seq dataset. We computed percent spliced in (PSI) for skipped exons (SE), mutually exclusive exons (MXE) and alternative 5′ and 3′ splice sites (A5SS and A3SS) as well as retained introns (RI)[31]. In total, 2% of detected AS events met our mis-splicing criteria (|ΔPSI| > 0.05, FDR < 0.05) (Fig. 3a). Among 866 changed AS events, SE was the most common outcome with mean 0.25 (0.05–1 range) |ΔPSI| value (Fig. 3b, c). Mis-splicing included MBNL1-regulated events that have been well documented in other tissues, including SE Clasp1 exon (E)20, A5SS Ncor2 E46 and MXE Dnm2 E10 and 11 (Fig. 3d and Supplementary Fig. 3). Moreover, 129-*Mbnl1* KO mice also showed striking splicing alterations in several genes, including transcription factors implicated in thymocyte/T-cell development such as Lef1 E6 and Tcf7 E5 as well as Tcf7 E10 (Fig. 3d, e)[32,33]. To support our computational splicing analysis, we performed RT-PCR assays on 129-*Mbnl1* KO and WT thymi as well as isolated thymocytes and confirmed significant changes in Lef1, Tcf7 and other transcripts (Fig. 3f and Supplementary Fig. 4a, b). Finally, we tested whether the AS events were also affected in B6-*Mbnl1* and B6.129-*Mbnl1* KO lines. We confirmed that thymic RNA misprocessing was independent of strain background since 129-*Mbnl1* and B6-*Mbnl1* congenics, as well as B6.129-*Mbnl1*

mixed background thymi, showed AS changes (Fig. 3g and Supplementary Fig. 4c, d). In agreement with low *Mbnl2* and *Mbnl3* expression levels in thymus, neither *Mbnl2*ΔE2/ΔE2 nor *Mbnl3*ΔE2/Y KOs showed a thymic splicing defect (Fig. 3g and Supplementary Fig. 4e, f).

To test if mouse *Lef1* E6 splicing was directly regulated by MBNL1, MBNL1-CLIP-seq clusters were identified near these SE events in publicly available data performed on mouse C2C12 myoblasts[34] (Fig. 3h). CLIP-seq reads formed a cluster located 21 nucleotides downstream of *Lef1* E6 and overlapped with nine YGCY/A (Y = pyrimidine) MBNL1 binding motifs predicted to be located mainly in bulges and stem-loops (Supplementary Fig. 4g, h). Compound knockdown of both Mbnl1 and Mbnl2 RNAs in C2C12 myoblasts resulted in Lef1 E6 skipping (Fig. 3h). To determine if MBNL proteins regulate splicing in human cells, MBNL paralog siRNA-mediated knockdown experiments were performed using human T lymphocyte (Jurkat) cells. As expected, MBNL1 siRNA knockdown also shifted splicing of human LEF1 E6 (Fig. 3i and Supplementary Fig. S4i). Finally, we performed similar analysis for Tcf7/TCF7 (Supplementary Fig. 4g and i, j). Therefore, MBNL1 loss in thymocytes caused pre-mRNA misprocessing and this developmental RNA processing factor directly regulated the alternative splicing of critical transcription factors required for normal T-cell development.

**Thymic mis-splicing replicated in peripheral blood cells.** The dysregulation of alternative splicing in *Mbnl1* KO thymus led us to test if similar RNA misprocessing events were detectable in

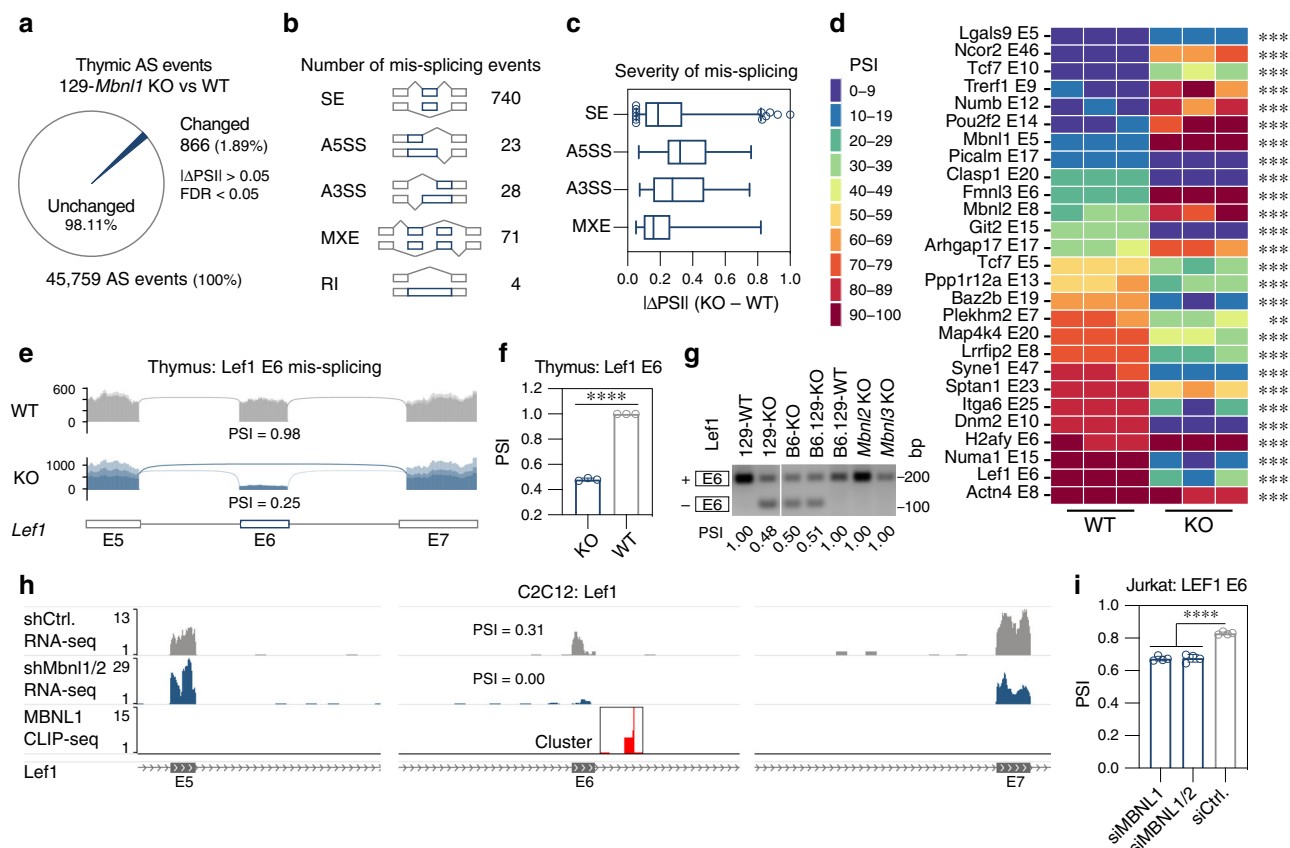

**Fig. 3 Alternative splicing changes in Mbnl1 KO thymus. a** Alternative splicing (AS) changes in 129-*Mbnl1* KO thymus. Pie chart represents the proportion of significantly altered AS events (blue) to all detected AS events (gray) in the 129-*Mbnl1* KO (n = 3) compared to WT (n = 3). **b** Number of significantly changed AS events categorized as skipped exons (SE), alternative 5′ and 3′ splice sites (A5SS and A3SS), mutually exclusive exons (MXE) and retained introns (RI). **c** Wide range of splicing changes. Box plot of AS event |ΔPSI| values. PSI—percent spliced in. Center line represents the median, and the box extends from the 25th to 75th percentiles. Whiskers show 1–99 percentile. **d** Heat map illustrates selected 27 AS events changed in *Mbnl1* KO thymus (compare to Fig. 5e). ** FDR < 0.01, **** FDR < 0.0001. **e** Sashimi plot of *Mbnl1* KO (n = 3) and WT (n = 3) RNA-seq samples for Lef1 E5-E6-E7. Mean PSI values are provided below the plots. **f** Lef1 E6 exclusion in 129-*Mbnl1* KO (n = 3) compared to WT (n = 3) thymi detected by RT-PCR. Bar graph shows mean PSI ± SD. Significant difference was determined by the two-tailed t-test: **** P < 0.0001. **g** Representative gel of Lef1 E6 RT-PCR assay from the different *Mbnl1* KO strains as well as *Mbnl2* and *Mbnl3* KOs. PSI values are provided below each gel lane. **h** Genome browser view of RNA-seq generated from C2C12 cells with the compound Mbnl1 and Mbnl2 knockdown by shRNA. The cluster of MBNL1 CLIP-seq reads indicates MBNL1 binding sites in C2C12. **i** MBNL1 (n = 4) as well as compound MBNL1 and MBNL2 (n = 4) knockdown causes LEF1 E6 exclusion in human T cells (Jurkat). Bar graph shows mean PSI ± SD. Significant difference was determined by Dunnett's multiple comparison test: **** adjusted P value ($P_{adj}$) < 0.0001. Source data are provided as the Supplementary Data 1 and Source Data files.

spleen, a secondary lymphoid and erythroid organ in the mouse[35]. We isolated spleens as well as splenocytes from 129-*Mbnl1* KO and WT mice and assessed Lef1 E6, Tcf7 E5 and Tcf7 E10 splicing by RT-PCR (Supplementary Fig. 5a, c). As expected, tested AS events showed the same mis-splicing pattern as detected in *Mbnl1* KO thymus. To determine if RNA mis-splicing identified in thymocytes and splenocytes might be detectable in blood lymphocytes, we harvested predominantly lymphoid in origin peripheral blood mononuclear cells (PBMCs) along with other DM-relevant tissues from B6.129-*Mbnl1* KOs and age-matched WT mice. Using RT-PCR, misprocessing of Lef1 E6 and Tcf7 E5, together with other previously reported MBNL-regulated AS events, was observed in PBMCs as well as other tissues (Fig. 4a, b and Supplementary Fig. 5d, e).

Recently, we identified CNBP CCUG expansion (CCUG$^{exp}$) induced intron 1 retention (CNBP-IR$^{exp}$) in DM2 tissues, including human PBMCs, and proposed this CNBP-IR$^{exp}$ event as an accessible and early-stage biomarker for DM2[36]. *CNBP*, unlike *DMPK*, is expressed at a relatively high level in mouse

thymus (Supplementary Fig. 1b) and human PBMCs where it is expressed 24-fold higher than *DMPK* (Fig. 4c and Supplementary Fig. 6a). The mis-splicing events detected in *Mbnl1* KO thymus and PBMCs motivated us to test whether MBNL1 sequestration by CCUG expansion mutations might cause similar AS changes in DM2 peripheral blood. Human PBMCs were isolated from DM2 blood (previously characterized in ref. [36]) with large >1000 CCTG repeats (DM2-LR) and short ~100 CCTG (DM2-SR) expansions together with DM1 and other controls followed by RNA-seq. Since DM2 and DM1 blood specimens were collected extramurally at DM patient conferences and shipped to our laboratory, the PBMC RNA-seq data revealed transcriptomic signatures characteristic of blood samples stored for prolonged periods, including previously reported changes in LEF1 E6 (Supplementary Fig. 6b-d)[37]. To overcome this difficulty, four MBNL1-dependent SE events were selected that were stable for at least 48 h at ambient temperature (Supplementary Fig. 6e)[37]. We observed significant AS changes in NCOR2 E47, MBNL1 E5 and two other transcripts only for DM2-LR, but not for DM2-SR,

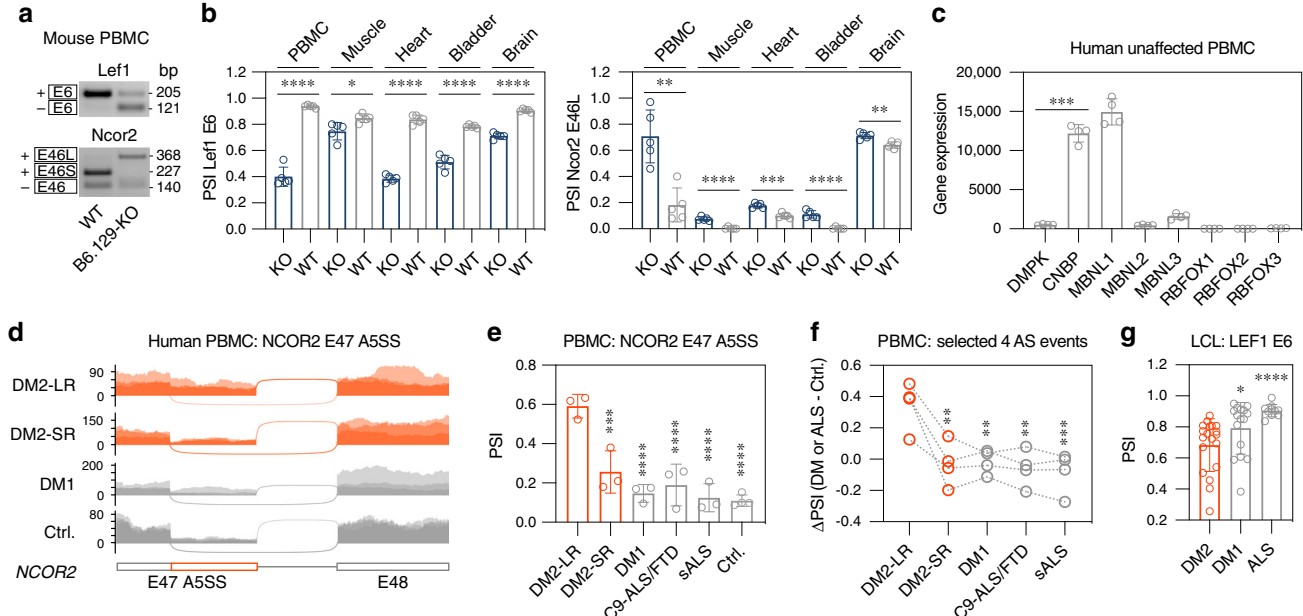

**Fig. 4 Alternative splicing changes in Mbnl1 KO and DM2 PBMCs. a** Representative RT-PCR assay gel of Lef1 E6 and Ncor2 E46 A5SS from 129.B6-*Mbnl1* KO and WT PBMCs. **b** 129.B6-*Mbnl1* KO ($n = 5$) causes Lef1 E6 exclusion and Ncor2 E46L inclusion in peripheral blood mononuclear cells (PBMCs), muscle (tibialis anterior), heart, bladder and brain. Bar graphs show mean PSI ± SD. Significant difference was determined by the two-tailed *t*-test test: *$P < 0.05$, **$P < 0.01$, ***$P < 0.001$, ****$P < 0.0001$. **c** Gene expression in unaffected human PBMCs. Bar graph shows mean expression value ± SD. Significant difference was determined by two-tailed *t*-test test: ***$P = 0.0003$. **d** Sashimi plot of DM2-LR (>1000 CCTGs; $n = 3$), DM2-SR (~100 CCTGs; $n = 3$), DM1 ($n = 3$) and unaffected control ($n = 4$; PBMC extraction delayed by 24–48 h) RNA-seq samples for NCOR2 E47-E48. **e** Aberrant NCOR2 E47L inclusion in DM2-LR but not in DM2-SR, DM1, C9-ALS/FTD nor sALS. Bar graph shows mean PSI ± SD. Significant difference was determined by Dunnett's multiple comparison test: ***$P_{adj} = 0.0004$, ****$P < 0.0001$. **f** Before-after plot shows individual ΔPSI values for NCOR2 E47, MBNL1 E5, SYNE1 E61 and GIT2 E16 AS events. Note that ΔPSI values for GIT2 E16 are inverted. Significant difference was determined by Dunnett's multiple comparisons test: **$P_{adj} < 0.01$, **$P_{adj} = 0.0009$. **g** Aberrant LEF1 E6 exclusion in a large cohort of DM2-derived LCLs ($n = 19$); DM1 ($n = 16$); ALS ($n = 10$). Significant difference was determined by Dunn's multiple comparison test: *$P_{adj} = 0.019$, ****$P_{adj} < 0.0001$. Source data are provided as the Supplementary Data 1 and Source Data files.

DM1, *C9orf72*-linked ALS/FTD (C9-ALS/FTD) and sporadic (s) ALS (Fig. 4d-f and Supplementary Fig. 6f). As expected, the severity of AS changes in DM2-LR and DM2-SR was associated with both CCTG$^{exp}$ size and CNBP-IR$^{exp}$ levels (Supplementary Fig. 6f-h).

Finally, we generated cDNA from a large cohort of lymphoblastoid cell lines (LCLs) from DM2, DM1, as well as disease and unaffected controls, and tested LEF1 E6 and SPTAN1 E23 splicing profiles by RT-PCR (Fig. 4f and Supplementary Fig. 6i). As expected, significant changes were only detected for DM2. Collectively, these results showed that MBNL1-dependent pre-mRNA misprocessing is detectable in thymocytes, splenocytes, PBMCs, LCLs, and STR expansion-induced spliceopathy was detected primarily in DM2 cells expressing CNBP RNAs with long CCUG repeat expansions. However, the CNBP-IR$^{exp}$ level in DM2-SR was elevated but still significantly lower compared to DM2-LR[36] (Supplementary Fig. 6h).

**Blood RNA splicing biomarkers for DM2**. To minimize the transcriptome artifacts induced by prolonged blood storage and/or processing, control and DM2 blood samples were directly collected into RNA/DNA preservation tubes (see Methods). All samples were screened for the CNBP-IR$^{exp}$ by RT-PCR and/or genomic blotting was performed to detect C(C)TG$^{exp}$ mutations (Fig. 5a). RNA-seq was performed using 6 samples for each group and CNBP-IR$^{exp}$ was detected in DM2 (Supplementary Fig. 7a). As expected, there was a strong correlation between CNBP-IR$^{exp}$

detection by RT-PCR and RNA-seq (Pearson's $r = 0.96$) (Supplementary Fig. 7b).

Since *CNBP*, unlike *DMPK*, is expressed at a relatively high level in human whole blood and we failed to detect splicing changes in DM1 PBMCs, the DM2 transcriptome was compared to both unaffected and DM1 controls (Supplementary Fig. 7c). Differential gene expression analysis revealed only ~0.2% of genes detected in human whole blood were mis-regulated (Supplementary Fig. 7d) and of the 58 genes that showed expression changes in DM2, 36 were upregulated while 22 were downregulated (Supplementary Fig. 7e). Similarly, <0.2% of detected AS events were altered in DM2 blood (Fig. 5b). Among 202 changed AS events, SE was the most common mis-splicing outcome with mean 0.19 (0.05–0.61 range) |ΔPSI| value (Fig. 5c, d). To identify the key splicing events in blood that would serve as reliable DM2 biomarkers, we selected 37 AS events based on parametric (gene expression, ΔPSI, FDR) and nonparametric (gene structure complexity, wiggle plot distribution) criteria (Fig. 5e and Supplementary Fig. 8). Importantly, 70% ($n = 26$) of AS events overlapped with our *Mbnl1* KO thymus results, including LEF1 E6 (Fig. 5e, f), and out of the 11 remaining AS events, 9 alternative exons were not identified in the mouse genome. For example, LGALS E5 is not present in mouse genome but was changed in Jurkat cells with MBNL1 knockdown, which indicated that these human AS exons are regulated by MBNL proteins (Supplementary Fig. 4e, f). Since whole blood AS events differ between patients, we selected 14 readily testable RT-PCR AS events and analyzed them in a large cohort of whole blood samples from DM2, DM1, sALS and unaffected controls.

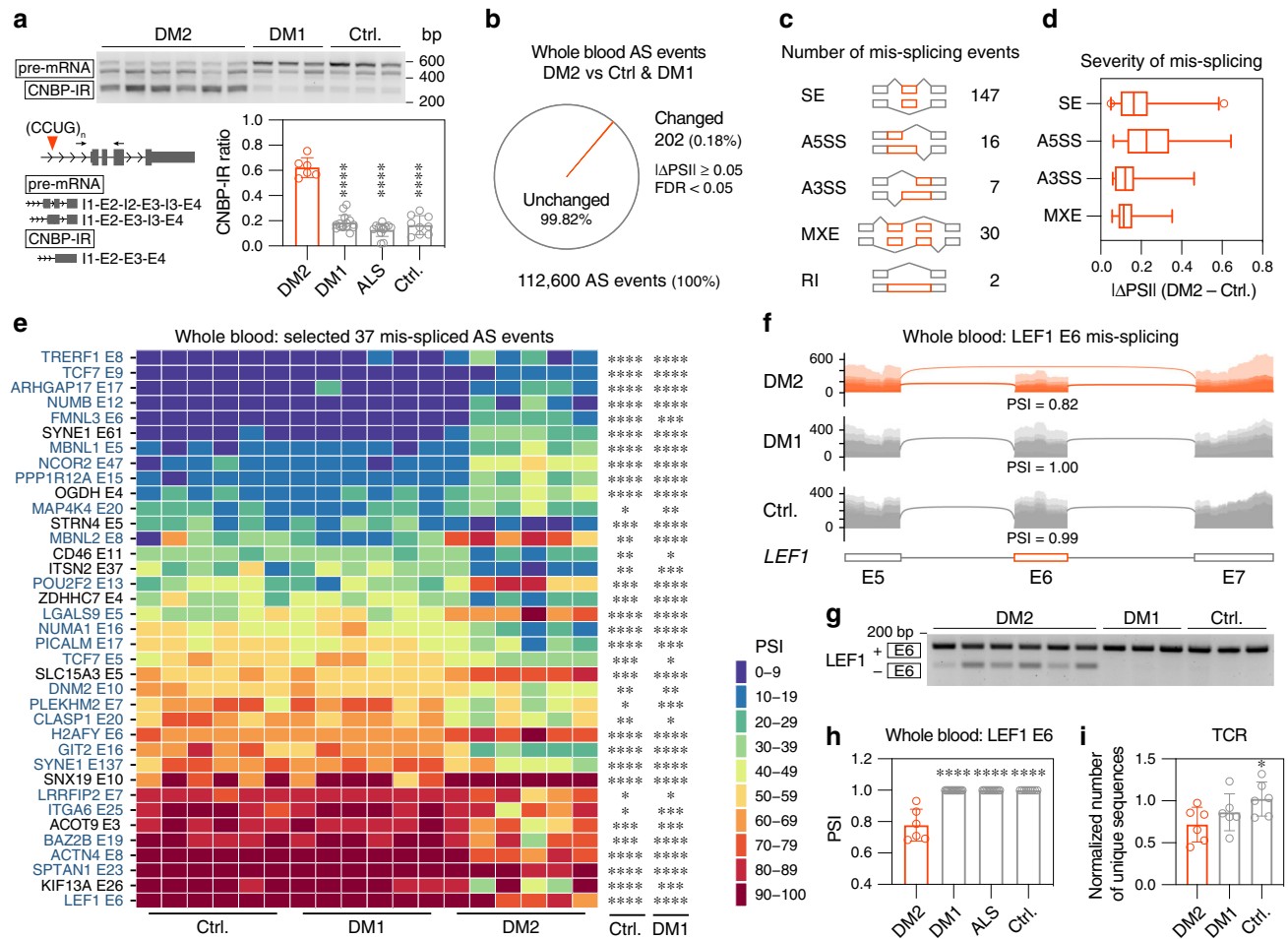

**Fig. 5 Alternative splicing changes in DM2 whole blood. a** RT-PCR analysis of CNBP-IR in DM2 ($n = 6$), DM1 ($n = 19$), ALS = ($n = 13$) and unaffected control ($n = 11$) whole blood samples. Bar graph shows mean CNBP intron 1 retention ratio ± SD. Significant difference was determined by Dunnett's multiple comparisons test: **** $P_{adj} < 0.0001$. **b** Pie chart represents all detected AS events in whole blood RNA-seq. Orange triangle represents significantly changed AS events in DM2 ($n = 6$) compared to DM1 ($n = 6$) and unaffected ($n = 6$) controls. **c** Number of significantly changed AS events assigned to different categories. **d** Box plot of AS event | ΔPSI | values. Center line represents the median, and the box extends from the 25th to 75th percentiles. Whiskers show 1-99 percentile. **e** Heat map shows selected 37 AS events changed in DM2 compared to both DM1 and unaffected controls. Note that TCF7 E9 is the mouse Tcf7 E10 ortholog. * FDR < 0.05, ** FDR < 0.01, *** FDR < 0.001, **** FDR < 0.0001. **f** Sashimi plot with mean PSI values for LEF1 E6 in DM2 ($n = 6$), DM1 ($n = 6$) and unaffected control (n = 6) RNA-seq samples. **g** Representative RT-PCR assay gel of LEF1 E6 from DM2, DM1 and unaffected control (Ctrl.) whole blood samples. **h** Aberrant LEF1 E6 exclusion in a large cohort of DM2 ($n = 6$), DM1 ($n = 19$), ALS ($n = 13$) and unaffected ($n = 11$) control whole blood samples. Significant difference was determined by Dunnett's multiple comparisons test: **** $P_{adj} < 0.0001$ **i** Reduction of TCR repertoire detected in DM2 whole blood RNA-seq. Bar graph shows the total number of TRB and TRA clones normalized to uniquely mapped reads ± SD. Significant difference was determined by Dunnett's multiple comparison test: * $P_{adj} = 0.042$. Source data are provided as the Supplementary Data 1 and Source Data files.

We demonstrated that these AS events are changed in DM2 and not in disease control samples (Fig. 5g, h and Supplementary Fig. 9a-e).

DM2 is a late-onset and progressive disease so we next examined if the AS changes detected in DM2 blood could be confounded by splicing defects observed during aging. We retrieved publicly available RNA-seq data from 7 younger (26–32-year old) and 7 older (94 year-old) adult female PBMCs[38]. Out of 1,333 age-associated splicing events only 8 overlap with 202 AS events detected in DM2 (Supplementary Fig. 10a-e). Only MAP4K4 E20 was significantly changed in these 37 selected AS events. Finally, we tested whether DM2-specific AS changes reflect a developmental delay in splicing pattern transitions from undifferentiated cells in bone marrow to differentiated cells in PBMCs. Out of 6209 developmentally associated events, 38 overlap with 202 AS events detected in DM2, including LEF 1 E6 (Supplementary Fig. 10f-j).

Since we noticed Tcrb and Tcra repertoire mis-regulation in Mbnl1 KO mice, we next assessed both TCR alpha and beta chain transcript repertoires in the whole blood RNA-seq dataset (Fig. 5i and Supplementary Fig. 9d). Our results indicated a reduced number of clones in DM2 compared to unaffected controls that was associated with the degree of spliceopathy (Supplementary Fig. 9e).

Based on these results, we concluded that specific mis-splicing outcomes provide blood-based biomarkers for DM2, and rapid analysis of whole blood samples presents a significant diagnostic advantage to detect this likely under-reported disease.

## Discussion

As thymocytes develop into functional T cells, approximately 90–95% are selectively deleted throughout the maturation process to ablate high affinity binding of self-antigens as well as promote recognition of non-self on antigen presenting cells (APCs)[39]. Transcriptional regulation has been shown to play a vital role in

thymic organogenesis and thymocyte development[40–42]. For example, immune tolerance requires negative selection mediated by medullary thymic epithelial cells (mTECs), which are characterized by promiscuous expression of tissue-specific antigens regulated by the *AIRE* gene and other factors to promote self-tolerance[43,44]. While AIRE facilitates global transcription for self-antigen expression, co-/post-transcriptional RNA alternative splicing during thymocyte education may also be crucial to create the diverse repertoire of self-epitopes that T cells encounter in the periphery. Surprisingly, only a few RNA splicing factors, including CELF2, HNRNPL, and SRSF2, have been implicated in thymic T cell development or thymopoiesis[5,6,45,46].

Here, we demonstrate that loss of the developmentally regulated RNA alternative splicing factor MBNL1 leads to dysregulated gene expression and RNA mis-splicing in the thymus resulting in mis-expression of important thymic regulatory factors. RNA-seq analysis uncovered several thymocyte/lymphocyte transcripts mis-spliced in *Mbnl1* KOs, including *Lef1 and Tcf7*. Both *Tcf7* and *Lef1* have been studied extensively in thymocyte development[32,33,47]. *Tcf7*[−/−] mice are characterized by disrupted thymocyte development, decreased thymic cellularity with decreased DP and SP thymocytes[48,49] in relatively young mice (6–8 weeks of age) similar to the cell population effects observed in HNRNPL-deficient thymus[5]. However, when *Tcf7*[−/−] mice are aged, 50% develop enlarged thymi[50]. By contrast, *Tcf7*[−/−]; *Lef1*[−/−] double KO (DKO) mice have no detectable defects in thymocyte development but they die shortly after birth[51]. Interestingly, *Tcf7; Lef1* DKO mice display increased T cell developmental disruption compared to single KO mice, suggesting functional redundancy between *Tcf1* and *Lef1* in thymic development[52]. Both Tcf7 and Lef1 alternative splicing is disrupted in *Mbnl1* KO thymus. The *Lef1* alternative exon altered by MBNL1 loss is located in the context dependent regulatory domain (CRD) and skipping of this exon reduces LEF1 binding affinity to the *Tcra* enhancer region resulting in decreased transcription and reduced TCR alpha chain[53]. Interestingly, prior studies have demonstrated that CELF2 promotes LEF1 E6 inclusion by interacting with sequences overlapping the MBNL1 CLIP-seq cluster (Supplementary Fig. 4g) indicating a possible compensatory role[6,53]. Additionally, *Tcf7* regulates transcription of beta catenin dependent transcripts and can serve as a negative regulator of *Lef1*[54,55]. The alternative exon included in *Mbnl1* KO mice encodes a c-clamp domain containing the 'CRARF' sequence that stabilizes binding between TCF7 and DNA[56,57]. The combined alternative splicing of both Tcf7 and Lef1 pre-mRNAs with the increased expression of *Lef1* suggests that TCR alpha chain production is compromised in the MBNL-deficient thymus.

DM1 and DM2 are both classified as myotonic dystrophies and a common pathomechanism for these diseases involves MBNL sequestration on C(C)UG[exp] RNAs. However, there are several molecular features that differentiate DM1 and DM2. While *CNBP* is expressed ubiquitously, and at a significantly higher level than *DMPK*, DM2 is generally considered less severe than DM1 with less muscle and brain RNA misprocessing. A potential reason for this discrepancy is that STR RNA toxicity is mitigated by RBFOX protein sequestration by CCUG[exp], but not CUG[exp], RNAs[58]. Interestingly, we demonstrated that this DM2 spliceopathy, other than CNBP-IR[exp], occurs with large, but not with relatively small (~100 CCTGs), DM2 CCTG expansions nor in DM1 blood cells. Our results indicate that similar to DM1, the extent of misprocessing in DM2 correlates with CCTG expansion size as well as CNBP-IR[exp] level (Supplementary Fig. 11).

Another important aspect of this study is that only a few RNA binding proteins involved in splicing, translation and RNA decay have been shown to play regulatory roles in T cell development and activation[45], so it is important to extend our knowledge of

co-/post-transcriptional regulation of the immune system. The relatively high level of *CNBP* and *MBNL1* expression in the thymus, and the inhibition of MBNL1 RNA processing activity in DM2 due to sequestration on CCUG[exp] RNAs, prompted us to determine if similar RNA mis-splicing events also occurred in DM2 PBMCs and whole blood. Our finding that specific RNA mis-splicing events in *Mbnl1* KO thymus are also detectable in DM2 whole blood and PBMCs indicates that human T cell development may also be altered by *CNBP* CCTG[exp] mutations. While the molecular basis of immune system dysfunction in DM patients is currently unclear, previous studies have shown a decrease in peripheral T cell populations in DM1 proportional to expansion length[59] and a high frequency of autoreactive T cell associated autoimmune disorders in DM2[18]. This work suggests that RNA mis-splicing in the immune system may be a characteristic feature of other RNA-mediated diseases, and further examination of additional microsatellite repeat expansion diseases that lead to loss of RBP functions may uncover immune deficits previously overlooked.

## Methods

***Mbnl1* knockout mice**. All relevant ethical regulations for animal testing and research were observed, and this study received ethical approval from the University of Florida Institutional Animal Care and Use Committee (IACUC). Mice were housed under specific pathogen free conditions. All animal procedures and endpoints were in accordance with IACUC guidelines and animals were sacrificed in accordance to IACUC approved protocols. B6.129S1-*Mbnl1*[ΔE3/ΔE3], *Mbnl2*[ΔE2/ΔE2] and *Mbnl3*[ΔE2/Y] have been described[60–62], and N10 congenic 129S1-*Mbnl1*[ΔE3/ΔE3] and B6-*Mbnl1*[ΔE3/ΔE3] lines were derived from B6.129S1-*Mbnl1*[ΔE3/ΔE3] mixed background mice.

**Immunoblotting**. Dissected tissues were homogenized in lysis buffer (20 mM HEPES-KOH, pH 8.0, 100 mM KCl, 0.1% Igepal CA-630 (Sigma), 0.5 mM phenylmethylsulphonyl fluoride, 5 μg/mL pepstatin A, 1 μg/mL chymostatin, 1 mM ε-aminocaproic acid, 1 mM p-aminobenzamidine, 1 μg/mL leupeptin, 2 μg/mL aprotinin) by disposable pestle followed by sonication on ice and centrifugation (16,100 × g, 15 min, 4°C). Protein lysates were quantitated using the DC Protein Assay (Bio-Rad) according to manufacturer's instructions. Total proteins (50 μg/lane) were resolved on 12.5% SDS-acrylamide and transferred to nitrocellulose membranes. Blots were blocked for 1 h in 5% non-fat dry milk in 1× PBS, 0.1% Tween-20 (Sigma). Primary antibodies, anti-MBNL1 A2764 (a gift from Dr. Charles Thornton, 1:5,000) and anti-GAPDH 6c5 (Abcam, ab8245, 1:10,000) were used for immunoblotting either overnight, 4 °C or 2 h, room temperature (RT). Blots were washed 3-times 1× PBS, 0.1% Tween-20 (Sigma), 10 min, RT prior to secondary antibody detection with anti-rabbit IgG or anti-mouse-IgG, conjugated to horseradish peroxidase (HRP) (GE Healthcare, NA934V and NA931V, 1:5,000) in wash buffer. Secondary antibodies were incubated 1 h, RT in wash buffer. Membranes were washed, as described above, and developed in Western Lightning Plus ECL detection reagents (Perkin Elmer) and exposed to Biomax Film (Kodak). Uncropped and unprocessed scans of the blots are provided in the Source Data file.

**Histology**. Paraformaldehyde (4%) fixed thymi were paraffin embedded using an automatic processor (VIP6, Leica) with graded ethanol (70–100%) and xylene by the University of Florida Molecular Pathology Core. Coronal sections (5 μm) were cut on a rotary microtome (Microm International). Sections were deparaffinized in xylene and rehydrated in graded ethanol (100-80%) prior to hematoxylin (RICCA Chemical) and eosin (Sigma) staining. Sections were dehydrated through graded ethanol (95–100%) and xylene prior to mounting with Cytoseal60 (Richard Allen Scientific). Histological tissue sections were scanned and saved as brightfield digital images using a whole slide scanner (Scanscope CS, Aperio, Leica, Vienna) and software (Aperio ImageScope 12.4).

**Thymocyte and splenocyte harvests**. Thymus and spleen were dissected into ice cold 1× PBS and connective tissue and fat removed. Single cell suspensions were made by mincing tissue through 100 μm cell strainers (BD Biosciences) into 1× PBS, pH 7.4. Cells were pelleted 400 × g, 10 min, RT and resuspended in 150 mM ammonium chloride, 1 mM potassium bicarbonate, 0.01 mM EDTA, pH 7.2 (Thermo Fisher) for red blood cell lysis. Cells were resuspended in RPMI-1640 (Mediatech) supplemented with 10% fetal bovine serum (FBS) (cRPMI; Hyclone). Viable cells were enumerated using Trypan Blue exclusion dye (Sigma).

**Flow cytometry**. Thymocyte composition was analyzed using anti-CD4 (RM4-5)-PE-Cy7 (1:200) or -APC (1:200) and anti-CD8 (53-6.7)-Pacific Blue (1:100) or

-PE-Cy5 (1:200) (BD Biosciences). Splenocytes were characterized using anti-CD3-Alexa700 (17A2, 1:200), anti-CD45/B220-Pacific Blue (RA3-6B2, 1:100), anti-CD4-PE-Cy7 (RM4-5, 1:200) and anti-CD8-PE-Cy5 (53-6.7, 1:200). Cells ($2 \times 10^6$) were stained in $1\times$ PBS, 0.5% BSA (FACS Buffer, Sigma) and analyzed using a BD LSR II (BD Biosciences). All cells analyzed were gated away from debris based on forward angle light scatter (FSC) and side angle light scatter (SSC) profiles and data were analyzed using FlowJo Data Analysis Software (Tree Star).

**Apoptosis analysis.** Thymocytes were plated in 24-well culture dishes (Corning) at $1.5 \times 10^6$ cells per well in cRPMI medium. Dexamethasone solubilized in DMSO (Sigma) was added to each well at final concentrations of $10^{-7}$ M and 0 M (vehicle only control) for each test sample. Cells were incubated 12 h at 37 °C, 5% CO$_2$ in a humidified chamber. Post treatment, cells were collected into 5 mL round bottom tubes and centrifuged $400 \times g$, 5 min, RT. Cells were resuspended in FACS Buffer and stained with anti-CD4 and anti-CD8 antibodies. Post incubation 30 min, 4 °C, light safe, cells were washed in FACS Buffer and centrifuged as above. Apoptotic markers Annexin V and propidium iodide staining were completed using FITC Annexin V Apoptosis Detection Kit I (BD Biosciences, 556547) according to manufacturer's protocol.

**Blood samples.** For blood isolation, mice (7 weeks of age) were terminally anesthetized with ketamine-xylazine based on the weight of the animal, then the posterior vena cava was severed, and blood collected by heparinized capillary (Fisherbrand) to BD Microtainer blood collection tubes with K$_2$EDTA (BD) and additional tissues were isolated for RNA extraction. All relevant ethical regulations for work with human participants were complied with and patient blood samples were collected following written informed consent as approved by the University of Florida Institutional Review Board (IRB). Sampling age is included in Supplementary Data 1. Blood was collected into DNA/RNA Shield Blood Collection Tubes (Zymo Research) or BD Vacutainer Tubes, ACD A (BD Biosciences). Peripheral blood mononuclear cells (PBMC) were isolated from the buffy coat of collected whole blood, followed by density gradient separation using Lympholyte-H according to manufacturer's provided protocol (Cedarlane). Red blood cells were lysed and removed by using the RBC Lysis Buffer (Roche). PBMCs were washed with PBS and used for either genomic DNA isolation (Flexigene kit; Qiagen), total RNA extraction (TRIzol).

**Cell culture and nucleofection.** Jurkat cells (American Type Culture Collection; ATCC) were cultured in RPMI 1640 medium with GlutaMAX (Gibco), supplemented with 10% fetal bovine serum (BioFluid) and 100 U/mL penicillin/streptomycin (ThermoFisher). Cells were cultured in water-jacketed incubators at 37 °C and 5% CO$_2$. Nucleofection of $2 \times 10^6$ cells were performed using Cell Line Nucleofector Kit V (Lonza) and Nucleofector II (Amaxa Biosystems) using program X-005. Jurkat cells were nucleofected with siRNA targeting MBNL1 (100 nM siMBNL1 and 100 nM control siRNA; Thermo Fisher Scientific), MBNL1 and MBNL2 (100 nM siMBNL1, 100 nM siMBNL2), or control siRNA (200 nM siRNA). Oligonucleotide sequences are included in Supplementary Data 1.

**RNA isolation.** Total RNA from DNA/RNA Shield Blood Collection Tubes was isolated by using Quick-DNA/RNA Blood Tube Kit with DNase treatment according to manufacturer's provided protocol (Zymo Research). Total RNA was isolated by using TRIzol Reagent and the Direct-zol RNA MiniPrep Kit with DNase treatment according to manufacturer's provided protocol (Zymo Research). Mouse tissues were homogenized in TRIzol (Ambion) with 1.5 mm zirconium beads in a Bead Ruptor 12 (OMNI International). Total RNA was isolated from thymus using PolyTron (Kinematica) homogenization in TRI Reagent (Sigma) followed by treatment with the RNeasy Kit (Qiagen) according to the manufacturer's instructions. RNA was quantitated on a Nanodrop, Quant-iT RiboGreen RNA Assay Kit (Thermo Fisher) and/or Qubit RNA BR Assay Kit (Thermo Fisher Scientific). RNA quality was assessed using a Bioanalyzer 2100 (Agilent Technologies).

**Splicing analysis by RT-PCR.** Total RNA was reverse transcribed using the GoScript Reverse Transcription System (Promega), SuperScript II or SuperScript III (Thermo Fisher Scientific) with random primers according to manufacturer's protocol. PCR was conducted using GoTaq G2 Flexi DNA Polymerase (Promega). PCR products were resolved on agarose gels stained with ethidium bromide and gels visualized on an ImageQuant 400 (GE Healthcare) or Molecular Imager ChemiDoc XRS + (BioRad) and analyzed using Multi Gauge software (Version 3.0; FujiFilm) or Image Lab (Version 6.0.1; BioRad). All primers and PCR product sizes are listed in Supplementary Data 1 (sheet 2). Original gel photos are provided in the Source Data file.

**RNA sequencing and computational analysis.** Thymic RNA-seq ($2 \times 50$ paired-end reads) were performed by The New York Genome Center from 9 weeks of age WT and 129S1-$Mbnl1^{\Delta E3/\Delta E3}$ thymic total RNA. For human PBMC and whole blood strand-specific, rRNA- and globin- (only whole blood) depleted RNA-seq libraries were prepared using the KAPA Stranded RNA-seq Kit with RiboErase

HMR Globin (Kapa Biosystems) per manufacturer's instructions, except for the use of custom Illumina-compatible index primers to allow multiplexing. Library size distribution was assessed using the High Sensitivity NGS Fragment Analysis Kit (DNF-747) on a Fragment Analyzer (Agilent) and $2 \times 76$ paired-end sequencing was performed using an Illumina NextSeq 500.

Publicly available RNA-seq data were retrieved from the Gene Expression Omnibus (GEO) database and are listed in Supplementary Data 1. Reads were aligned to the human hg38 or mouse mm10 genomes. Salmon[63] was used for transcript expression quantification and differential gene expression analysis was performed using DESeq2[64]. MiXCR was used for repertoire analysis and Shannon's equitability was computed[65,66]. For splicing analysis, reads were aligned using STAR[67] followed by rMATS[31] (Version 4) analysis. The IGV browser[68] was used for data visualization and ggsashimi[69] was used for Sashimi plot generation. CLIP-seq data were derived from DMseq.org[70].

**Statistical information and data visualization.** Whole transcriptome statistical analysis for gene expression and alternative splicing were performed using DEseq2 and rMATS, respectively. Other statistical analyses, including survival analysis, were performed by using GraphPad Prism software (Version 8). The normal distribution was assessed by the Shapiro–Wilk test followed by parametric or non-parametric tests. For multiple comparison one-way ANOVA was followed post hoc test. Details are specified in the figure legends. Graphs where generated in GraphPad Prism software. Heat maps were generated in R using the ggplot2 package.

**Reporting summary.** Further information on research design is available in the Nature Research Reporting Summary linked to this article.

## Data availability
The GEO accession numbers for all RNA-seq data used in this study are listed in Supplementary Data 1 file (sheet 3 and 4). RNA-seq data generated during this study have been deposited in GEO under accession GSE138691. The source data underlying all Figures and Supplementary figures are provided as the Source Data and Supplementary Data 1 files. All data and biological materials are available from the corresponding author upon reasonable request.

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

## Acknowledgements

The authors thank the patients and their families for their contributions to this study and members of the Center for NeuroGenetics for technical assistance. We would like to thank A. Cowsert and J. D. Cleary for help with blood collection and T. M. Brusko for comments on the manuscript. This work was funded by grants from the National Institute of Health (NIH NS058901, NS098819 to M.S.S and L.P.W.R.), the Muscular Dystrophy Association (MDA546770 to Ł.J.S; MDA480539 to M.S.S.), the University of Florida McKnight Brain Institute (to Ł.J.S.), the Polish National Science Centre (2017/24/C/NZ1/00112 to K.T.), and the Myotonic Dystrophy Foundation (to C.A.N.).

## Author contributions

Ł.J.S, M.S.S., and M.M.S. designed the research; Ł.J.S, M.M.S., J.S., K.T., F.I., and C.A.N. performed the experiments; Ł.J.S, M.M.S., J.S., K.T., F.A., and M.S.S. analyzed the data; S.H.S. and L.P.W.R contributed human samples; Ł.J.S. performed the data visualization; Ł.J.S., M.M.S., and M.S.S. wrote the manuscript.

## Competing interests

M.S.S. is a member of the scientific advisory boards of Locana Bio and Skyhawk Therapeutics. All other authors declare no competing interests.
