## [Peer Review File · Nature Communications]

Reviewers' comments:

Reviewer #1 (Remarks to the Author):

The authors investigate the role of the RNA binding protein, MBNL in thymus development and adaptive immunity. This is a new area of investigation. Their results show that Mbnl1 KO mice have thymic hyperplasia and accumulation of thymocytes after birth. Molecular analysis by RNA-seq identifies alternative splicing misregulation of genes important for T cell maturation. Importantly, the splicing changes are identified in the blood of individuals affected by myotonic dystrophy type 2, a disease in which MBNL activity is reduced due to sequestration on mutant RNA containing CUG repeat expansions.

RNA-seq of 9wk old mice showed misregulation of ~1400 genes including strong down regulation of immunoglobulin heavy and kappa genes. Alternative splicing alterations were validated by RT-PCR and correlations with Mbnl1 binding established using previously generated CLIP data in C2C12 myoblasts. Splicing abnormalities were also detected in peripheral blood mononuclear cells. Interestingly, the authors found that the gene containing an expanded CCTG and causing myotonic dystrophy type 2 (DM2) was highly expressed in PBMCs. This suggests that in DM2, loss of MBNL1 function by sequestration on the CCUG repeats could show the same changes as loss of Mbnl1. Selected splicing events in DM2 PBMCs were shown to reflect splicing changes consistent with MBNL1 loss of function. The authors make a cogent argument that these splicing changes could serve as a blood biomarker for DM2 although many more samples will need to be tested

Comments

What is the significance of the changes in cell populations presented in Fig 2f?

The level of reduced apoptosis does not appear to be enough to explain the hypercellularity overall. But there appears to be large variability in the effects on thymus hypertrophy in individual animals. It would be useful to have more information on the variability of thymus hypertrophy. It seems that for some animals it is dramatic and others barely noticeable. Is this the case and does the variability correlate with other factors such as Mbnl2 compensation or differences in histology/effects on different cell types?

A caveat with the results in Fig 4d-f is the relatively small number of examples and selective nature of the splicing events shown. A more robust overlap of DM2 vs MBNL1-dependent splicing event changes are presented in Fig 5 so the point of Fig 4d-f is not clear.

Since the "aging" analysis includes what seems to be only one 94 year old female, the results are more anecdotal than useful

Reviewer #2 (Remarks to the Author):

In this manuscript the authors, who are widely recognized experts in studies of MBNL deficiency and RNA splicing, present the first analysis of the lymphoid phenotype in MBNL1 knockout mice. They

previously showed that MBNL1 is very highly expressed in thymus, and now show that MBNL1 KO causes substantial thymic hyperplasia (possibly due to defective apoptotic signaling), partial inhibition of thymic maturation, and abnormal splicing of many cassette exons including in thymic transcription factors Lef6 and Tcf7. An important strength of the paper is the transcriptome data they generated from a unique and valuable resource, peripheral blood cells from DM1 and DM2 patients. Ultimately they define a set of splicing events that may provide novel biomarkers for DM2 in peripheral blood, thus expanding their earlier report that CNBP intron retention might serve as a marker. Since the human DM2 PBMCs express high levels of the MBNL1-sequestering CNBPexp transcript, it is satisfying to observe some of the same splicing defects that were detected in MBNL1-KO mouse lymphocytes. The study is well done and I have only relatively minor comments and questions.

1. Do the authors have any insight as to why the hyperplasia is heterogeneous such that roughly a quarter of the male mice and half of the females don't show the phenotype?
2. Assays of Lef1 splicing in spleen were performed on splenocytes, but spleen is not only a lymphoid organ, but also a secondary erythroid organ in the mouse. Nevertheless, the authors' conclusion that Lef1 splicing is abnormal in MBNL1-deficient splenic lymphocytes is almost certainly correct, because any erythroid progenitors remaining after red cell lysis presumably aren't expressing Lef1. However, assays of non-lymphoid specific genes would also be interrogating erythroid cells.
3. The paper suggests that "spliceopathy occurs in DM2, but not DM1, blood cells and the extent of mis-processing correlates with CCTG expansion size possibly due to low expression of RBFOX proteins in blood." Can the authors comment more specifically on the expression of RBFOX proteins in lymphocytes? There is some data showing low expression in human PBMCs in Figure 4c but otherwise it's not reported.
4. Figure 3c – Sorry, I don't quite understand the graph here or in Figure 5d or Supplementary Figure 10c, d. Naively one would expect that percent inclusion and percent exclusion would correlate precisely for a given exon, but obviously I'm not interpreting this correctly. Is it actually two datasets combined into one figure?
5. Major changes in alternative splicing were detected in Lef1 and perhaps to a lesser extent in Tcf7, transcription factors implicated in T cell development. The authors discuss some possible defects expected from disrupted transcription, such as decreased expression of TCR alpha chain, but they do not actually say whether they observed decreased TCR alpha chain expression in the MBNL1 knockout mice. Is this observed?
6. p. 11: CNBP intron 1 retention was not exclusively detected in DM2, rather it was at a significantly higher level in DM2.
7. Figure 5a: It's clear that CNBP-IR was quite different in DM2 samples compared to controls, but not at all clear exactly what was measured in this figure. Maybe a diagram to show what regions are being

amplified would clarify, either here or in the supplement.

8. I did not find PCR primer sequences in the Supplementary Information.

Minor comments:

1. For non-hematologists, it would be nice to explicitly mention that PBMCs are largely lymphoid in origin.

2. It would also be helpful to indicate consistently which samples are human.

3. p. 10: Please define LCL (presumably, lymphoblastoid cell lines).

4. Figure 3d: the number of significance designations (asterisks) does not match the number of genes shown in the heatmap.

5. Figure 4f: Perhaps I misunderstand the Y axis, but if the data represents the absolute value of PSI differences, why are there negative values? Also, the text says that cDNA from a “large cohort” of DM2, DM1, and control cell lines was analyzed. How many cell lines of each type were analyzed? (looks like 4 of each in the figure?)

6. Figure S3 and S4a,e,f: Some figures refer to Tcf7 E9 while others refer to Tcf7 E10. Is this a labeling error, or are both exons affected?

7. Supplementary Figure 4c: it’s difficult to see any difference between wt and ko for Tcf7 in the RNA-seq plots.

8. Figure S6. Both RNA-seq and cDNA analyses are described on p. 10. It would be great to clarify which type of data is represented in Supplementary Figure 6f.

Reviewer #3 (Remarks to the Author):

This manuscript by Swanson and colleagues identifies a set of splicing events that are common in PBMCs of DM2 patients and may serve as biomarkers of this disease. They come to this conclusion by studying the impact of Mbnl1 in thymocytes and the lymphoid lineage. Mbnl1 is functionally depleted in DM1 and DM2 via sequestration, and patients with these diseases present with hallmarks of thymic dysregulation. Consistently, in their mouse model of Mbnl1 genetic depletion, the authors observe dysregulation of thymic cellularity and dysregulation of gene expression. These same changes in gene expression, including several splicing patterns, are preserved in mature PBMCs and can be observed in whole blood. While overall this study is interesting and informative, there are omissions in much of the

phenotypic and mechanistic parts of the study that weaken the final conclusions. It is important to fill in at least many of these gaps to strengthen the manuscript.

Major comments:

1) The authors make a big point of the changing cellularity and size of the thymus in figure 2, but the reasons for this are not pursued. Several easy experiments should be done to better document if, where and why a block in development exists: (a) Is *Mnbl* expressed equally in DN, DP and SP cells, or does it just change in a subset? (b) Is *Mnbl1* just expressed in the thymocytes or also in TECs? (c) why show thymic composition in the complicated way used in Fig 2f rather than the typical reporting of percentages of each population in FACS plots such as in 2e? And also the authors should report numbers of TECs (which could be driving the increased thymic size). (d) is the reduction in apoptosis observed in all populations (i.e. CD4 SP, CD8 SP, DP, DN) or only a subset (which might then account for the relative loss of CD4 vs CD8 SP)? (e) is there a skewing of CD4s to CD8s in peripheral blood? (f) do any of the splicing or gene expression changes observed in the KO cells account directly for the reduction in apoptosis?

2) The authors focus on altered splicing of LEF1 exon 6 in the *Mbnl1* KO mice as an important functional target and a hallmark of disease. This splicing change is a clear phenotype, but is also expected given the work from the Lynch lab that has shown a requirement for the YGCY/A motifs downstream of LEF1 exon 6 in promoting inclusion of this exon, and has already shown that LEF1 exon 6 inclusion varies throughout thymic development (Mallory et al., MCB 2011; PNAS 2015). In addition, this previous work has shown that CELF2 (which binds to similar sequences as *Mbnl1*), binds to the regulatory sequence downstream of LEF1 exon 6, varies in expression in developing thymocytes, and drives changes in splicing in developing thymocytes including promoting inclusion of LEF1 exon 6. In this context, it is important for the authors to ask if *Mbnl1* functions cooperatively with CELF2, or maybe these two proteins are expressed at distinct stages of development and compensate for other another. This is important for the final conclusions of the manuscript, as the fact that LEF1 exon 6 has been shown to be regulated by other proteins (CELF2) and pathways (JNK signaling), suggests that dysregulation of this splicing event might occur disease states (such as cancer) that were not compared to DM2 in Figure 5.

Minor comments:

1) The plots in 3c and 5d are unclear. Each SE data point has both an exclusion and inclusion value, but these don't equal to 100. How can an exon be included 40% and excluded 10%. What is the other 50%?

2) The thresholds for calling splicing changes (5% deltaPSI) are lower than the typical 10% used in most other studies. The events the authors focus on are changing much more than this threshold, so it is unclear what the advantage is of being so permissive and how much of this is false positives.

3) The square for the 22 wk KO thymus in panel 2a is slightly blown up (based on scale lines compared to the boxes above and to the left) and is bigger than the others. The thymus is obviously bigger, but this comparison would look more appropriate if the same size and scale images were used.

Response to Reviewers

We would like to thank the Reviewers for their insightful and constructive comments on our manuscript (NCOMMS-19-539531) entitled 'Loss of MBNL1 induces mis-processing in the thymus and peripheral blood' submitted to *Nature Communications*. As detailed below, we have responded to all of the your concerns and suggestions for improvement and modified the text (see changes marked in blue font) and/or figures accordingly while also conforming to the character limit. Overall, we believe the Reviewers' comments and our responses have resulted in a significantly improved manuscript.

Reviewer #1

The authors investigate the role of the RNA binding protein, MBNL in thymus development and adaptive immunity. This is a new area of investigation. Their results show that Mbnl1 KO mice have thymic hyperplasia and accumulation of thymocytes after birth. Molecular analysis by RNA-seq identifies alternative splicing misregulation of genes important for T cell maturation. Importantly, the splicing changes are identified in the blood of individuals affected by myotonic dystrophy type 2, a disease in which MBNL activity is reduced due to sequestration on mutant RNA containing CUG repeat expansions. RNA-seq of 9wk old mice showed misregulation of ~1400 genes including strong down regulation of immunoglobulin heavy and kappa genes. Alternative splicing alterations were validated by RT-PCR and correlations with Mbnl1 binding established using previously generated CLIP data in C2C12 myoblasts. Splicing abnormalities were also detected in peripheral blood mononuclear cells. Interestingly, the authors found that the gene containing an expanded CCTG and causing myotonic dystrophy type 2 (DM2) was highly expressed in PBMCs. This suggests that in DM2, loss of MBNL1 function by sequestration on the CCUG repeats could show the same changes as loss of Mbnl1. Selected splicing events in DM2 PBMCs were shown to reflect splicing changes consistent with MBNL1 loss of function. The authors make a cogent argument that these splicing changes could serve as a blood biomarker for DM2 although many more samples will need to be tested.

We thank the Reviewer for these comments. Sample size selection for DM2 RNA-seq experiments was determined by power analysis. For DM2, we analyzed a total of 31 blood or blood-derived samples, including whole blood (n = 6), PBMCs (n = 6) and LCLs (n = 19). Importantly, many of the mis-regulated AS events detected in DM2 blood have been reported in biopsied DM1 and DM2 skeletal muscles (Nakamori et al., 2013; Wagner et al, 2016) although in this study we introduce new AS biomarkers (e.g., LEF1 E6).

Major:

1. What is the significance of the changes in cell populations presented in Fig 2f?

Now we demonstrate that DN/CD8⁺ accumulation and the reduction of DP are caused by a developmental delay or block at the DN/ISP stage. We have now added new information and data (Revised Fig. 2g and Supplementary Fig. 2c-e).

2. The level of reduced apoptosis does not appear to be enough to explain the hypercellularity overall. But there appears to be large variability in the effects on thymus hypertrophy in individual animals.

The Reviewer is correct that apoptosis may not explain all of the hypercellularity and we have now added a sentence (p. 9) that emphasizes this point. We also provided gene ontology results supporting disrupted apoptotic proliferation pathways (p. 8 and 9).

3. *It would be useful to have more information on the variability of thymus hypertrophy. It seems that for some animals it is dramatic and others barely noticeable. Is this the case and does the variability correlate with other factors such as Mbnl2 compensation or differences in histology/effects on different cell types?*

Yes, we observed heterogeneity in the enlarged thymus phenotype as indicated in Fig. 2b and Supplementary Fig. 2a. In response to the Reviewer's comment, we now provide more information and data on thymic phenotype variability (p. 7-9) and Fig. 1f and Supplementary Figs. 1h and 2d. The Reviewer brings up the interesting suggestion that MBNL2 compensation in some mice may explain this variable phenotype and we noted an increase in Mbnl2 at the RNA level in *Mbnl1* KO mice even in non-enlarged thymi (Supplementary Fig. 1d).

4. *A caveat with the results in Fig 4d-f is the relatively small number of examples and selective nature of the splicing events shown. A more robust overlap of DM2 vs MBNL1-dependent splicing event changes are presented in Fig 5 so the point of Fig 4d-f is not clear.*

The major point of Fig. 4d-f is to demonstrate that small, although generally considered to be pathogenic, CTG expansions do not cause alternative splicing changes in PBMCs (other than an elevated CNBP-IR^{exp}). In our opinion, it is important to point out that AS changes, similar to DM1, depends on expansion size, and we have now emphasized this point in the Discussion (p. 18). Unfortunately, pre-symptomatic DM2 samples with very small expansions are difficult to obtain, and thus we were unable to collect additional samples.

5. *Since the "aging" analysis includes what seems to be only one 94 year old female, the results are more anecdotal than useful.*

We have now included additional samples (7 94-year old adult females and 7 26-32 years old female controls), and have added a sentence (p. 14) that clarifies this issue.

Reviewer #2

*In this manuscript the authors, who are widely recognized experts in studies of MBNL deficiency and RNA splicing, present the first analysis of the lymphoid phenotype in MBNL1 knockout mice. They previously showed that MBNL1 is very highly expressed in thymus, and now show that MBNL1 KO causes substantial thymic hyperplasia (possibly due to defective apoptotic signaling), partial inhibition of thymic maturation, and abnormal splicing of many cassette exons including in thymic transcription factors *Lef6* and *Tcf7*. An important strength of the paper is the transcriptome data they generated from a unique and valuable resource, peripheral blood cells from DM1 and DM2 patients. Ultimately they define a set of splicing events that may provide novel biomarkers for DM2 in peripheral blood, thus expanding their earlier report that CNBP intron retention might serve as a marker. Since the human DM2 PBMCs express high levels of the MBNL1-sequestering CNBP^{exp} transcript, it is satisfying to observe some of the same splicing defects that were*

detected in MBNL1-KO mouse lymphocytes. The study is well done and I have only relatively minor comments and questions.

We thank the Reviewer and appreciate these supportive comments.

Major:

1. *Do the authors have any insight as to why the hyperplasia is heterogeneous such that roughly a quarter of the male mice and half of the females don't show the phenotype?*

As noted in the response to Reviewer 1 (R1), we observed heterogeneity in the enlarged thymus phenotype as indicated in Fig. 2b and for this revision we now provide data on thymic phenotype variability (p. 7-9.) and in the revised Figs. 1 and 2 and Supplementary Figs. 1 and 2. We know that factors such as age, sex and mouse strain background impact the penetrance of the thymic hyperplasia in *Mbnl1* KO mice.

2. *Assays of Lef1 splicing in spleen were performed on splenocytes, but spleen is not only a lymphoid organ, but also a secondary erythroid organ in the mouse. Nevertheless, the authors' conclusion that Lef1 splicing is abnormal in MBNL1-deficient splenic lymphocytes is almost certainly correct, because any erythroid progenitors remaining after red cell lysis presumably aren't expressing Lef1. However, assays of non-lymphoid specific genes would also be interrogating erythroid cells.*

The Reviewer is correct that splicing changes can be detected in other splenocyte subpopulations and we have modified the text to address this issue (p. 11)

3. *The paper suggests that "spliceopathy occurs in DM2, but not DM1, blood cells and the extent of mis-processing correlates with CCTG expansion size possibly due to low expression of RBFOX proteins in blood." Can the authors comment more specifically on the expression of RBFOX proteins in lymphocytes? There is some data showing low expression in human PBMCs in Figure 4c but otherwise it's not reported.*

We regret not providing the data to fully support our statement, and have deleted "possibly due to low expression of RBFOX proteins in blood" from this sentence (p. 18).

4. *Figure 3c – Sorry, I don't quite understand the graph here or in Figure 5d or Supplementary Figure 10c, d. Naively one would expect that percent inclusion and percent exclusion would correlate precisely for a given exon, but obviously I'm not interpreting this correctly. Is it actually two datasets combined into one figure?*

The Reviewer is correct and to address this concern we have replaced the original scatter plots with box plots that better demonstrate the degree of mis-splicing (revised Figs. 3c and 5d and Supplementary Fig. 10c and h).

5. *Major changes in alternative splicing were detected in Lef1 and perhaps to a lesser extent in Tcf7, transcription factors implicated in T cell development. The authors discuss some possible defects expected from disrupted transcription, such as decreased expression of TCR alpha chain, but they do not actually say whether they observed decreased TCR alpha chain expression in the MBNL1 knockout mice. Is this observed?*

To address this concern, we now provide data to demonstrate that the TCR repertoire is altered in both 129-*Mbn1* KO and DM2 whole blood (Fig. 1e-g; Fig. 5i; Supplementary Fig. 1g-h; Supplementary Fig. 9 d-e).

6. *p. 11: CNBP intron 1 retention was not exclusively detected in DM2, rather it was at a significantly higher level in DM2.*

We have modified the text to address this issue (p.13).

7. *Figure 5a: It's clear that CNBP-IR was quite different in DM2 samples compared to controls, but not at all clear exactly what was measured in this figure. Maybe a diagram to show what regions are being amplified would clarify, either here or in the supplement.*

We now include the RT-PCR strategy as a part of the revised Fig. 5a.

8. *I did not find PCR primer sequences in the Supplementary Information.*

The PCR primer sequences are included in Supplementary Data (Spreadsheet 2), and we have now modified the tab name (Primers; siRNA) to highlight this issue.

Minor:

1. *For non-hematologists, it would be nice to explicitly mention that PBMCs are largely lymphoid in origin.*

We now mention that PBMCs are primarily lymphoid in origin (p. 11).

2. *It would also be helpful to indicate consistently which samples are human.*

In addition to blue and orange colors that we use for *Mbn1* KO and DM2, respectively, we have now added additional graph labels. We have also modified the text to address this issue.

3. *p. 10: Please define LCL (presumably, lymphoblastoid cell lines).*

The Reviewer is correct and LCL is now defined when used initially (p. 12).

4. *Figure 3d: the number of significance designations (asterisks) does not match the number of genes shown in the heatmap.*

We thank the Reviewer for catching this error which has been corrected.

5. *Figure 4f: Perhaps I misunderstand the Y axis, but if the data represents the absolute value of PSI differences, why are there negative values? Also, the text says that cDNA from a "large cohort" of DM2, DM1, and control cell lines was analyzed. How many cell lines of each type were analyzed? (looks like 4 of each in the figure?).*

Fig. 4f has been corrected (not absolute Δ PSI) and we note (p. 36, Fig. 4f legend) that Δ PSI values for GIT2 E16 are inverted. In our studies, we analyzed a total of 31 blood or

blood derived DM2 samples, including whole blood (n = 6), PBMCs (n = 6) and LCLs (n = 19). A "large cohort" relates to the LCLs in Fig. 4g.

6. *Figure S3 and S4a,e,f: Some figures refer to Tcf7 E9 while others refer to Tcf7 E10. Is this a labeling error, or are both exons affected?*

In mouse we examined Tcf7 E5 and E10 (Supplementary Figs. 3 and 4a,e,f) and for human the orthologous exon for mouse Tcf E10 is TCF E9 Fig. S8).

7. *Supplementary Figure 4c: it's difficult to see any difference between wt and ko for Tcf7 in the RNA-seq plots.*

We acknowledge that this difference is difficult to visualize in the browser view due to low Tcf7 expression in C2C12, however we were able to quantify Tcf7 E5 inclusion level using rMATS (revised Supplementary Fig. 4g).

8. *Figure S6. Both RNA-seq and cDNA analyses are described on p. 10. It would be great to clarify which type of data is represented in Supplementary Figure 6f.*

Supplementary Figure 6f represents human PBMC RNA-seq results, and we have modified the Fig. 6f legend to address this issue.

Reviewer #3

This manuscript by Swanson and colleagues identifies a set of splicing events that are common in PBMCs of DM2 patients and may serve as biomarkers of this disease. They come to this conclusion by studying the impact of Mbnl1 in thymocytes and the lymphoid lineage. Mbnl1 is functionally depleted in DM1 and DM2 via sequestration, and patients with these diseases present with hallmarks of thymic dysregulation. Consistently, in their mouse model of Mbnl1 genetic depletion, the authors observe dysregulation of thymic cellularity and dysregulation of gene expression. These same changes in gene expression, including several splicing patterns, are preserved in mature PBMCs and can be observed in whole blood. While overall this study is interesting and informative, there are omissions in much of the phenotypic and mechanistic parts of the study that weaken the final conclusions. It is important to fill in at least many of these gaps to strengthen the manuscript.

We agree with the Reviewer and have attempted to fill these gaps in this revision.

Major:

1. *The authors make a big point of the changing cellularity and size of the thymus in figure 2, but the reasons for this are not pursued. Several easy experiments should be done to better document if, where and why a block in development exists: (a) Is Mbnl1 expressed equally in DN, DP and SP cells, or does it just change in a subset? (b) Is Mbnl1 just expressed in the thymocytes or also in TECs? (c) why show thymic composition in the complicated way used in Fig 2f rather than the typical reporting of percentages of each population in FACS plots such as in 2e? And also the authors should report numbers of TECs (which could be driving the increased thymic size). (d) is the reduction in apoptosis observed in all populations (i.e. CD4 SP, CD8 SP, DP, DN) or only a subset (which might then account for the relative loss of CD4 vs CD8 SP)? (e) is there a skewing of CD4s to*

CD8s in peripheral blood? (f) do any of the splicing or gene expression changes observed in the KO cells account directly for the reduction in apoptosis?

We provided more information and data on thymic phenotype variability (p. 7-9) and in revised Fig. 2 and Supplementary Fig. 2.

(a, b) We now provide additional data on *Mbnl1* expression in ETP, DN2a, DN2b, DN3a, DN3b, DN4, ISP, DP, CD4⁺, CD8⁺, and mTEC (Supplementary Fig. 2b).

(c) We exchanged the original plots with simple bar plots that more clearly demonstrate percent changes in thymocyte subpopulations (Fig. 2f). Although we are not able to report the number of TECs, we noticed a predominant thymocyte overgrowth based on H+E staining.

(d) We appreciate the Reviewer's question but have not been able to perform the suggested dexamethasone-induced apoptosis experiment since this would require the generation of additional KO mice which would exceed the resubmission window.

(e) We provided additional data on splenocyte (as a model of the periphery) composition in 129-*Mbnl1* KO (Supplementary Fig. 2f-g).

(f) We now include gene ontology results demonstrating that 124 differentially expressed genes are involved in apoptotic processes (Supplementary Data; Spreadsheet 21).

2. *The authors focus on altered splicing of LEF1 exon 6 in the Mbnl1 KO mice as an important functional target and a hallmark of disease. This splicing change is a clear phenotype, but is also expected given the work from the Lynch lab that has shown a requirement for the YGCY/A motifs downstream of LEF1 exon 6 in promoting inclusion of this exon, and has already shown that LEF1 exon 6 inclusion varies throughout thymic development (Mallory et al., MCB 2011; PNAS 2015). In addition, this previous work has shown that CELF2 (which binds to similar sequences as Mbnl1), binds to the regulatory sequence downstream of LEF1 exon 6, varies in expression in developing thymocytes, and drives changes in splicing in developing thymocytes including promoting inclusion of LEF1 exon 6. In this context, it is important for the authors to ask if Mbnl1 functions cooperatively with CELF2, or maybe these two proteins are expressed at distinct stages of development and compensate for other another. This is important for the final conclusions of the manuscript, as the fact that LEF1 exon 6 has been shown to be regulated by other proteins (CELF2) and pathways (JNK signaling), suggests that dysregulation of this splicing event might occur disease states (such as cancer) that were not compared to DM2 in Figure 5.*

We thank the Reviewer and further acknowledge the prior important studies of Mallory et al on CELF2. For this revision, we have modified the text (p. 17) and figures (revised Supplementary Fig. 4g).

Minor:

1. *The plots in 3c and 5d are unclear. Each SE data point has both an exclusion and inclusion value, but these don't equal to 100. How can an exon be included and excluded 10%. What is the other 50%?*

The Reviewer is correct, and we have replaced the original figures with simple box plots that better highlight the degree of mis-splicing (revised Figs. 3c, 5d and Supplementary Fig. 10c,h).

2. *The thresholds for calling splicing changes (5% deltaPSI) are lower than the typical 10% used in most other studies. The events the authors focus on are changing much more than this threshold, so it is unclear what the advantage is of being so permissive and how much of this is false positives.*

Based on our false discovery rate (FDR) threshold, we anticipate no more than 5% of type I errors (false positive results). Since the splicing changes in DM2 whole blood are relatively mild, we selected the lower 5% Δ PSII, and were able to confirm AS changes with Δ PSII < 10% by RT-PCR (e.g. SYNE1 E61; Supplementary Fig. 9b-c).

3. *The square for the 22 wk KO thymus in panel 2a is slightly blown up (based on scale lines compared to the boxes above and to the left) and is bigger than the others. The thymus is obviously bigger, but this comparison would look more appropriate if the same size and scale images were used.*

We have adjusted Fig. 2a according to the Reviewer's suggestion.

We hope the Reviewers will find this revised submission suitable for *Nature Communications*.

Sincerely yours,

Maurice S. Swanson
Professor

REVIEWERS' COMMENTS:

Reviewer #1 (Remarks to the Author):

The authors have addressed my concerns

Reviewer #2 (Remarks to the Author):

In this revised manuscript, the manuscript is improved by addressing previous critiques and by addition of new information. The authors have mostly addressed my original concerns (but see #1 below). A few very minor issues were identified in the new material as listed below (#2-4)

1. Original minor point #8 had requested clarification of Figure S6f. The response says that the legend was modified to address the issue, but the legend appears unchanged to me.
2. The new results analyzing CD4+ / CD8+ cells and gene expression provide more details about the effects of MBNL1 loss on thymocyte differentiation. However, there is an apparent discrepancy in the legend for Fig. S2e where it says CD4 gene expression increases in the KO thymus, whereas the Figure and main text say that CD4 expression is reduced in the KO thymus. Also, it appears that CD4 expression is increased in splenocytes (Fig. S2g) – is this a real difference compared with thymocytes? How might this occur?
3. p. 9 "...24% of 129-MBNL1 KOs did not develop an abnormal thymic phenotype by ~22 age of age" – two things: first, it was 24% of males that did not develop an abnormal phenotype; and second, it should be "weeks" of age.
4. Fig. 2g: Yes, we can clearly see differences in the FSC profile for WT vs KO in the older mice, but how does this indicate a developmental delay before the DP stage as mentioned on p.8?

Reviewer #3 (Remarks to the Author):

The authors have addressed all my primary concerns with additional data and text.

We would like to thank the Reviewers for their helpful comments on our manuscript (NCOMMS-19-539531A) entitled 'Loss of *MBNL1* induces mis-processing in the thymus and peripheral blood' to *Nature Communications*. As detailed below, we have responded to the remaining concerns of Reviewer #2 and modified the text accordingly (track changes feature in MS Word).

Reviewer #1

The authors have addressed my concerns.

Reviewer #2

In this revised manuscript, the manuscript is improved by addressing previous critiques and by addition of new information. The authors have mostly addressed my original concerns (but see #1 below). A few very minor issues were identified in the new material as listed below (#2-4).

We thank the Reviewer and appreciate these comments.

1. *Original minor point #8 had requested clarification of Figure S6f. The response says that the legend was modified to address the issue, but the legend appears unchanged to me.*

The Reviewer is correct. Previously, we added "PBMC: transcript and E number" on the top of bar graph (Supplementary Figure 6f). Now, we have modified the legend: "f Aberrant *MBNL1* E5, *SYNE1* E61 and *GIT2* E16 splicing in DM2-LR but not in DM2-SR, DM1, C9-ALS/FTD nor sALS RNA-seq PBMC samples. Bar graphs show mean PSI \pm SD. Significant difference were determined by Dunnett's multiple comparisons test: # $P_{adj} = 0.157$, ** $P_{adj} < 0.01$, *** $P_{adj} < 0.001$ " (p.17).

2. *The new results analyzing $CD4^+/CD8^+$ cells and gene expression provide more details about the effects of *MBNL1* loss on thymocyte differentiation. However, there is an apparent discrepancy in the legend for Fig. S2e where it says *CD4* gene expression increases in the KO thymus, whereas the Figure and main text say that *CD4* expression is reduced in the KO thymus.*

We thank the Reviewer for catching this error, and we have corrected the Supplementary Figure 2e description. "f *Cd4* and *Notch1* gene expression decrease and increases in *Mbnl1* KO (n = 3) compared to WT (n = 3) thymus, respectively. Bar graph shows mean expression \pm SD. DESeq2; * $P_{adj} < 0.05$ " (p. 6).

*Also, it appears that *CD4* expression is increased in splenocytes (Fig. S2g) – is this a real difference compared with thymocytes? How might this occur?*

The Reviewer is correct. The CD4⁺ subpopulation seems to be overrepresented in 129-Mbn1 KO spleen. Since we anticipate the CD4⁺/CD8⁺ ratio is altered in 129-*Mbn1* KO we modified the text (p. 8). "Finally, to extend our 129-Mbn1 KO characterization, we analyzed peripheral T cells in spleen, and noted an age-related change in the CD4⁺/CD8⁺ ratio, possibly due to thymic dysfunction (Supplementary Fig. 2g-h)" (p. 8).

3. p. 9 "...24% of 129-MBNL1 KOs did not develop an abnormal thymic phenotype by ~22 age of age" – two things: first, it was 24% of males that did not develop an abnormal phenotype; and second, it should be "weeks" of age.

We thank the Reviewer for catching our omissions, and we have modified the text to address these issues. "Although gene expression changes were detectable as early as 9 weeks of age when thymi were in the normal size range, 42% of 129-Mbn1 KOs did not develop enlarged thymi. Factors as age, sex and genetic strain background impacted the penetrance of this phenotype" (p. 9).

4. Fig. 2g: Yes, we can clearly see differences in the FSC profile for WT vs KO in the older mice, but how does this indicate a developmental delay before the DP stage as mentioned on p.8?

We thank the Reviewer for this comment. To address this issue, we have modified the text and provided an additional explanation. "Since ISP are larger than DP and mature CD8⁺ thymocytes²⁶, we analyzed the size distribution of the CD8⁺ subpopulation. The forward angle light scatter (FSC) analysis revealed an age-related increase in relative cell size distribution in 129-Mbn1 KO compare to WT indicating aberrant accumulation of ISP cells (Fig. 2g and Supplementary Fig. 2e)" (p. 8).

Reviewer #3

The authors have addressed all my primary concerns with additional data and text.

We hope this modified revised submission is ready for final acceptance for publication in *Nature Communications*.

Sincerely yours,

Maurice S. Swanson
Professor